# Exploring the Role of Neuropeptide PACAP in Cytoskeletal Function Using Spectroscopic Methods

**DOI:** 10.3390/ijms25158063

**Published:** 2024-07-24

**Authors:** Roland Gábor Vékony, Andrea Tamás, András Lukács, Zoltán Ujfalusi, Dénes Lőrinczy, Veronika Takács-Kollár, Péter Bukovics

**Affiliations:** 1Department of Biophysics, Medical School, University of Pécs, 7624 Pécs, Hungary; rvkony35@gmail.com (R.G.V.); andras.lukacs@aok.pte.hu (A.L.); zoltan.ujfalusi@aok.pte.hu (Z.U.); denes.lorinczy@aok.pte.hu (D.L.); veronika.kollar@aok.pte.hu (V.T.-K.); 2Department of Anatomy, Medical School, University of Pécs, 7624 Pécs, Hungary; andreatamassz@gmail.com

**Keywords:** PACAP, actin, fluorescence spectroscopy, cytoskeletal rearrangement, biomarker, TBI

## Abstract

The behavior and presence of actin-regulating proteins are characteristic of various clinical diseases. Changes in these proteins significantly impact the cytoskeletal and regenerative processes underlying pathological changes. Pituitary adenylate cyclase-activating polypeptide (PACAP), a cytoprotective neuropeptide abundant in the nervous system and endocrine organs, plays a key role in neuron differentiation and migration by influencing actin. This study aims to elucidate the role of PACAP as an actin-regulating polypeptide, its effect on actin filament formation, and the underlying regulatory mechanisms. We examined PACAP27, PACAP38, and PACAP6-38, measuring their binding to actin monomers via fluorescence spectroscopy and steady-state anisotropy. Functional polymerization tests were used to track changes in fluorescent intensity over time. Unlike PACAP27, PACAP38 and PACAP6-38 significantly reduced the fluorescence emission of Alexa488-labeled actin monomers and increased their anisotropy, showing nearly identical dissociation equilibrium constants. PACAP27 showed weak binding to globular actin (G-actin), while PACAP38 and PACAP6-38 exhibited robust interactions. PACAP27 did not affect actin polymerization, but PACAP38 and PACAP6-38 accelerated actin incorporation kinetics. Fluorescence quenching experiments confirmed structural changes upon PACAP binding; however, all studied PACAP fragments exhibited the same effect. Our findings indicate that PACAP38 and PACAP6-38 strongly bind to G-actin and significantly influence actin polymerization. Further studies are needed to fully understand the biological significance of these interactions.

## 1. Introduction

Pituitary adenylate cyclase-activating polypeptide (PACAP) was first isolated from the sheep hypothalamus in 1989 [1,2]. It belongs to the vasoactive intestinal peptide (VIP)/glucagon/secretin peptide family, sharing 68% structural homology with VIP, but stimulating adenylate cyclase activity 1000 times more effectively in rat pituitary cells [2]. PACAP plays crucial roles in the nervous and endocrine systems, acting as a hormone, neurotransmitter, vasodilator, and neuromodulator, and potentially having neurotrophic or neuroprotective functions [1,2]. PACAP exists in two biologically active forms, PACAP38 and PACAP27, and our research also includes the artificial fragment PACAP6-38 [3,4]. PACAP is encoded by the *ADCYAP1* gene, located on chromosome 18 [5]. This gene produces multiple isoforms of PACAP through alternative splicing, including PACAP38 and PACAP27.

PACAP receptors are linked to the G-protein S subtype (G_S_ protein), exhibiting stimulatory effects [6]. PACAP can bind to VPAC1 (VIP receptor type 1) and VPAC2 (VIP receptor type 1) receptors, which also bind VIP, due to their structural similarity. However, the PACAP Receptor 1 (PAC1 receptor) is PACAP-specific, binding strongly to both natural forms and the artificial fragment PACAP6-38, but with low affinity to VIP [7,8]. The high specificity of PAC1 receptors for PACAP makes them ideal for studying PACAP–receptor interactions. PAC1 receptors, being G_S_ protein-coupled receptors (GPCRs), induce stimulatory effects through secondary messengers upon binding PACAP [9,10].

PACAP protects various peripheral organs, including the lungs, liver, kidneys, gastrointestinal tract, heart, pancreas, and skin, and may affect bone/joint diseases, like rheumatism or osteoarthritis [11,12,13,14,15,16]. It has vasodilatory, antioxidative, and cytoprotective effects, reducing inflammation, apoptosis [17], and oxidative stress [18,19,20,21,22,23]. PACAP is involved in insulin secretion regulation [24,25,26,27,28] and promotes bone and cartilage formation [29,30]. Its presence in the skin influences sweat gland activity and protects against inflammatory conditions [23,31,32,33,34,35,36]. PACAP, implicated in migraine development, has an unclear mechanism. However, blocking PAC1 receptors or PACAP38 could provide a novel treatment approach, similar to targeting CGRP, which shares receptor characteristics [37,38]. PACAP also protects retinal pigment epithelial cells under hyperosmotic and oxidative stress, potentially offering therapeutic benefits for macular edema and diabetic retinopathy [39]. The multifaceted roles of PACAP highlight its therapeutic potential across multiple organ systems.

PACAP synergizes with multiple factors to promote neurite growth in different cell lines [40,41,42]. It works with nerve growth factor (NGF) and VIP to stimulate neuritogenesis, increasing neurite length and branching, similar to brain-derived neurotrophic factor (BDNF). PACAP enhances axon length and cell body size but does not affect dendrite length. PACAP6-38 inhibits PACAP-induced axon growth and also affects BDNF-induced growth [40,41,42]. In traumatic central nervous system (CNS) injuries, PACAP has cytoprotective and antiapoptotic effects. It increases dihydropyrimidinase like 2 (DPYL2) protein levels in ischemic brain tissue, marking axonal and neural growth, particularly in the early stages [43,44]. Animal models of traumatic brain injury (TBI) show that intracerebroventricular (icv) PACAP reduces axonal swellings and amyloid plaque accumulation in brain regions such as the corticospinal tract and medial longitudinal fasciculus [45,46]. In our previous study [47], we found that, in patients with severe head injury (Glasgow Coma Scale [GCS] ≤ 8), the endogenous PACAP levels in plasma rise significantly on the second day post-TBI, correlating with one-week mortality [47]. The protective effects of PACAP in optic nerve, facial nerve injuries, and TBI support its potential as a therapeutic agent for both peripheral and central nervous system injuries [46,47,48].

The polymerization of G-actin into filamentous actin (F-actin) is essential for maintaining cellular structural integrity and supporting dynamic processes. Actin filaments, which form the cell’s cytoskeleton, provide mechanical strength and stability. The F-actin network supports various cellular functions, including cell migration, division, and intracellular transport, by continuously polymerizing and depolymerizing [49,50]. The dynamic nature of F-actin allows cells to adapt to environmental changes and respond to signals, interacting with regulatory proteins to influence membrane protrusions, cell adhesion, and polarity [51,52]. PACAP significantly impacts axonal growth by necessitating actin polymerization [53,54]. PACAP influences the cytoskeletal system, particularly in cerebellar granule cells, by opposing C2-ceramide’s effects on cell motility and neurite growth through differential Tau protein phosphorylation, actin distribution, and tubulin polymerization [55]. The cytoskeleton, comprising microtubules (tubulin), microfilaments (actin), and intermediate filaments, facilitates intracellular transport, cell morphology, and mechanical protection [56]. PACAP, similar to forskolin, promotes actomyosin stress fiber depolymerization by increasing cAMP levels, impacting astrocyte growth via the phosphoinositide-3-kinase and RhoA protein pathways [57].

PACAP has a strong neurotrophic and neuroprotective effect, which has been investigated in many studies [58,59,60], and affects Schwann cell functions and myelin maturation [61], and PACAP receptors are upregulated after peripheral nerve injury [62]. PACAP promotes axonal development [63], increases axonal sprouting [64], and induces neuronal differentiation in growing and regenerated neurons [65,66,67].

Neuroprotective effects have been demonstrated in various neurodegenerative diseases and neuronal insults, including spinal atrophy [68], amyotrophic lateral sclerosis [69], Alzheimer’s disease [70], stroke [71,72], Parkinson’s disease [58,59,73,74], and Huntington chorea [75].

A recent study on PACAP and neuropathy in a streptozotocin-induced diabetes model [76] found that PACAP reduced morphological signs of nerve injury (axon–myelin separation, elevated mitochondrial number in myelinated axons, unmyelinated fiber atrophy, and basement membrane thickening of endoneurial vessels) while also attenuating pressure and touch hyperalgesia.

Several animal and human studies have reported reduced PACAP levels, altered PACAP receptor expression, and the beneficial effect of PACAP on cognition and memory in conditions associated with cognitive deficits, such as aging, Alzheimer’s, Parkinson’s, and Huntington’s disease, schizophrenia, and fragile X syndrome [77,78].

As mentioned above, PACAP plays a significant role in mitigating neuronal damage and promoting cell survival. Although PACAP itself does not typically misfold or aggregate, it influences the pathology of neurodegenerative diseases through its regulatory role in the cytoskeleton and interaction with proteins like Tau. The misfolding and aggregation of proteins such as Tau are hallmarks of neurodegenerative diseases like Alzheimer’s [70]. PACAP enhances Tau phosphorylation and prevents its dephosphorylation, potentially protecting against the formation of neurofibrillary tangles, which are characteristic of Alzheimer’s disease. Additionally, PACAP inhibits caspase-3 activity, reducing Tau cleavage and possibly mitigating Tau aggregation and its associated cytoskeletal disruptions [1,79]. By counteracting the effects of C2-ceramide, which reduces Tau phosphorylation and promotes its degradation via caspase-3, PACAP supports cytoskeletal integrity and neuron protection [55]. Thus, modulatory effects of PACAP on Tau and other cytoskeletal components suggest a significant protective role against neurodegenerative processes associated with protein misfolding and aggregation [1,55,70,79,80].

In previous investigations [81,82], we employed differential scanning calorimetry (DSC) to explore the impact of various PACAP fragments on the thermal stability of Ca^2+^-G actin and Ca^2+^-F actin. Our findings revealed that full-length PACAP (PACAP38) and its fragment PACAP27 significantly influence the thermal properties of actin, suggesting a complex interplay between these molecules. PACAP38 induced a notable shift in the denaturation temperature and introduced exothermic transitions in the higher temperature range, indicating structural modifications and interactions within the actin filaments. The PACAP27 fragment displayed distinct thermal behavior with higher calorimetric enthalpy, implying more extensive conformational changes compared with PACAP38. Other PACAP fragments, such as PACAP6-38 and PACAP6-27, also altered the thermal stability of actin, albeit to a lesser degree. These DSC results underscored the intricate thermodynamic interactions between PACAP and actin subunits (very probably between subdomain 1 and 3 in G-actin), providing insights into their potential roles in modulating actin filament stability and dynamics [81,82].

In the present study, we aimed to investigate the role of PACAP in cytoskeletal regulation using various in vitro methods, including steady-state anisotropy, fluorescence emission measurements, polymerization tests, and quenching studies. These approaches were intended to provide insights into the binding affinities of different PACAP forms to globular actin (G-actin) and their effects on actin polymerization, a crucial process for the dynamic functions of the cytoskeleton, such as intracellular transport and structural support. Actin polymerization involves nucleation, elongation, and treadmilling, with the critical concentration (C_C_) and equilibrium dissociation constant (K_D_) indicating polymer stability. Given the limited in vitro studies on PACAP as a potential actin-regulating polypeptide, our research aimed to elucidate the indirect role of PACAP in cytoskeletal regulation, understand the underlying regulatory mechanisms and interactions, and investigate the impact of PACAP on actin monomer incorporation into filaments. We also conducted quenching experiments to examine the fluorescence decrease emitted by the intrinsic fluorophores of actin upon interaction with PACAP, characterizing both static and dynamic quenching properties. To achieve these objectives, we intended to conduct absorption and emission spectroscopy measurements using PACAP38, PACAP27, and PACAP6-38 sequences (Figure 1).

The cryo-electron microscopy structure of the human PAC1 receptor in complex with PACAP38 was described in 2020 [83]. Figure 2 illustrates the 3D structure of PACAP27, which is identical to the first 27 residues of PACAP38, highlighting key aromatic residues involved in fluorescence (green: tyrosine; red: phenylalanine). This structural representation visualizes the spatial arrangement of PACAP residues critical for their roles in fluorescence.

Additionally, our goal was to explore the therapeutic potential of PACAP by understanding its extensive direct and indirect effects, building on previous research findings. This comprehensive approach aimed to advance our knowledge of the role of PACAP in cytoskeletal regulation and its potential clinical applications.

## 2. Results

The results obtained from temporally separated measurements exhibited consistent trends, with standard deviation values remaining within the normal range, indicating the reliability of the methods used. The following sections present our findings, illustrated with corresponding figures.

### 2.1. Steady-State Anisotropy and Fluorescence Emission

We examined the anisotropy and fluorescence emissions of A488NHS-labeled G-actin (0.2 µM) in the presence of increasing PACAP concentrations (0–1.5035 μM) (+1 mM EGTA). Fitting the data yielded nearly identical dissociation equilibrium constants (K_D_) (100–150 nM) for PACAP38 and PACAP6-38, whereas PACAP27 showed very weak interactions (Figure 3). The two-way analysis of variance (ANOVA) indicated significant differences in binding to actin monomers between PACAP forms (*p* < 0.001). The post hoc Tukey HSD test confirmed significant differences between PACAP38 and PACAP27 (*p* < 0.001), and PACAP6-38 and PACAP27 (*p* < 0.05), with no significant difference between PACAP38 and PACAP6-38. These findings suggest significant binding affinities of PACAP38 and PACAP6-38 to G-actin, while PACAP27 exhibited minimal interaction. No significant change was observed in the anisotropy of G-actin in the presence of PACAP27, consistent with fluorescence emission intensity changes.

Our results indicate distinct binding behaviors of different PACAP fragments to G-actin. Figure 4 shows that increasing concentrations of PACAP38 (Figure 4A) and PACAP6-38 (Figure 4C) decreased the detected actin monomer intensity. A similar, though not identical, trend with a different pattern was observed with PACAP27 (Figure 4B).

From the spectra, we derived the F/F_0_ intensity curves (Figure 5). The areas under the 30 nm spectra corresponding to maximum intensity values were integrated and plotted as relative values against PACAP concentrations. The ratios of control data and actin emission values corresponding to the relevant PACAP concentrations were used. The two-way ANOVA showed significant differences between PACAP forms (*p* < 0.05). The post hoc Tukey HSD test approached significance between PACAP38 and PACAP27 (*p* = 0.0575). The decreasing intensity may indicate conformational changes in G-actin, suggesting potential interactions as the fluorophore becomes less exposed to the solvent.

### 2.2. Actin Polymerization Assay

The actin polymerization assay was conducted to further investigate the effects of PACAP forms on actin (Figure 6). Pyrene-labeled actin (2.5 μM, 5% labeled) was used to monitor the actin polymerization kinetics in the presence of varying concentrations of PACAP forms (0–12.5 μM). Our results indicate that PACAP27 does not significantly affect actin polymerization, whereas PACAP38 and PACAP6-38 substantially alter the kinetics of actin incorporation. At low concentration ranges (nM to a few µM), they inhibit actin monomer incorporation, with the elongation rate decreasing by up to 20% at approximately 3 µM PACAP. At the highest applied concentration (12.50 µM), the relative polymerization rate for PACAP38 returned to its initial value. Notably, PACAP6-38 showed an approximately 50% increase in polymerization rate, indicating a biphasic nature.

The results indicate a concentration-dependent effect of PACAP on actin polymerization rates. These observations suggest that PACAP38 inhibits polymerization up to around 3 µM, likely sequestering actin monomers at lower concentrations. In contrast, PACAP27 does not significantly impact the kinetics of actin monomer incorporation.

### 2.3. Fluorescence Quenching Study

The fluorescence quenching experiments confirmed that the binding of PACAP induces structural change in actin. Figure 7 depicts the quenching of tryptophan fluorescence of actin by acrylamide in the presence or absence of various PACAP forms/fragments. The calculated Stern–Volmer Constants (K_SV_) do not suggest a greater change in the structure of actin, but the binding of PACAP induced a conformational change with no doubt. Interestingly, all three types of PACAP fragments exerted the same effect, proving that the extent of conformational change was independent of the PACAP isoform. Our experiments revealed that the binding of PACAP to actin filaments moderately decreased the accessibility of tryptophan residues. Unlike previous studies, PACAP27 showed similar quenching effects to PACAP38 and PACAP6-38, suggesting that the absence of one tyrosine did not significantly impact quenching.

## 3. Discussion

### 3.1. Cytoskeletal Considerations

The polymerization of G-actin into filamentous actin (F-actin) is crucial for maintaining cellular structural integrity and supporting dynamic processes. Actin filaments form the cell’s cytoskeleton, providing mechanical strength and stability. The F-actin network supports various cellular functions, including cell migration, division, and intracellular transport, by continuously polymerizing and depolymerizing [49,50]. The dynamic nature of F-actin allows cells to adapt to environmental changes and respond to signals, interacting with regulatory proteins to influence membrane protrusions, cell adhesion, and polarity [51,52]. PACAP significantly impacts axonal growth, necessitating actin polymerization [53,54].

Our research focused on the effect of PACAP on actin dynamics to better understand its role in cellular processes. PACAP, a neuropeptide involved in various physiological processes, influences cytoskeletal dynamics, crucial for axon regeneration and injury responses. Abnormal cytoskeletal dynamics in injured axons can form dystrophic structures, inhibiting regeneration. Proper pharmacological modification can transform these structures into growth cones, aiding recovery [84,85]. Given the known effects of PACAP, it shows promise as a potential agent for neural damage and other nervous system pathologies. Our findings explore these predictions further.

### 3.2. Interpretation of the Results

Our in vitro experiments support in vivo studies, enhancing our understanding of the effects of PACAP in various pathological conditions. We confirmed differences in PACAP binding to G-actin among the different PACAP forms, likely due to variations in binding mechanisms and structural differences.

PACAP38 and PACAP6-38, differing by only five amino acids, show similar binding strengths to G-actin, as indicated by the high anisotropy values. Despite PACAP6-38 being an artificial and physiologically degraded fragment, its three-dimensional structure is likely similar to that of PACAP38 [86]. Both forms demonstrated strong binding, with no significant difference in their dissociation equilibrium constants. Conversely, anisotropy values indicated that PACAP27 exhibited significantly weaker binding. This suggests that the lower affinity of PACAP27 is due to the absence of the last 11 amino acids (28th to 38th), which are crucial for interaction with G-actin. The weaker binding of PACAP27 implies lower efficacy in therapeutic applications compared with PACAP38 and PACAP6-38. Given that PACAP38 is more prevalent in the human body, it is likely to play a key role in peripheral organ effects. Further research is needed to fully understand PACAP27’s potential applications or to refute its therapeutic relevance. Despite the limited efficacy of PACAP27, other unexamined fragments might yield results similar to PACAP38 or PACAP6-38, warranting further investigation. The significant difference in interaction is likely due to the absence of the last 11 amino acids in PACAP27.

The differences in receptor binding highlight the research potential of PACAP38 and PACAP6-38. Future studies, including total internal reflection fluorescence (TIRF) microscopy, could provide a more accurate understanding of the structure and effects of PACAP. Determining the exact structure will clarify many unanswered questions and advance PACAP-related in vitro studies. The impact of PACAP on neurite number and growth, facilitated by actin polymerization, suggests potential treatments for neurologically derived degenerative diseases [41,42], for central nervous system disorders [43,44,45,46], severe head injuries [47], and regenerative medicine [87]. PACAP38 and PACAP6-38 may enhance this process, while PACAP27’s limited results suggest a minimal contribution compared with the other forms. Current experimental methods should investigate additional PACAP fragments to identify potentially more effective forms.

The emission spectra results aligned with anisotropy measurements, confirming PACAP27’s distinct behavior. Unlike PACAP38 and PACAP6-38, PACAP27 did not show the expected decrease in actin monomer intensity with increasing concentrations, likely due to structural differences. Although PACAP27’s intensity values were similar to the other forms, they lacked a consistent pattern, suggesting limited therapeutic potential. In contrast, PACAP38 and PACAP6-38 showed a clear decrease in intensity with increasing concentration, indicating conformational changes in G-actin. The decrease in fluorescence intensity suggests that PACAP38 and PACAP6-38 interact with G-actin, causing structural changes that reduce fluorophore exposure. This indicates stronger interactions compared with PACAP27, as confirmed by both anisotropy and emission measurements.

PACAP, primarily found in the central nervous system, impacts axon growth, nerve cell regeneration, and other neuronal functions [3,4]. The polymerization tests revealed that PACAP38 and PACAP6-38 significantly influence actin filament polymerization, essential for cytoskeleton regeneration. PACAP27 showed a minimal impact, correlating with its weaker interaction with G-actin. PACAP38 and PACAP6-38 positively affected polymerization rates, especially at higher concentrations, with PACAP6-38 being the most effective. PACAP27 had negligible effects, aligning with other measurement outcomes. Considering the polymerization tests, actin filament polymerization is crucial for cytoskeletal restoration. Both PACAP38 and PACAP6-38 demonstrated excellent results, suggesting therapeutic potential for various diseases.

Our quenching studies showed that, though PACAP27 had one less tyrosine than the other two isoforms, they all exerted the same effect on actin filaments, indicating that the absence of one tyrosine did not significantly affect interaction and the caused change in structure. As all four tryptophan residues are located in subdomain 1 of the actin molecule, our results suggest that the binding of PACAP affects the conformation of actin with its binding on this site. Further studies with extrinsic fluorophores bound to certain residues of actin subdomain 1 would reveal a more accurate picture of the discovered structural change.

Our spectroscopic analysis revealed that PACAP38 and PACAP6-38 exhibited strong binding to G-actin, significantly enhancing the polymerization rate, while PACAP27 showed a weaker interaction. This is consistent with our findings from differential scanning calorimetry (DSC), where PACAP38 and PACAP6-38 decreased the denaturation temperature of Ca^2+^-F-actin compared with the control, indicating reduced thermal stability, while PACAP27 and PACAP6-27 did not show significant changes [81,82]. The quenching studies showed both static and dynamic properties across all PACAP forms, aligning with the observed binding affinities. Specifically, the high affinity and significant calorimetric enthalpy changes in PACAP38 and PACAP6-38 mixtures [82] suggest a pronounced interaction with actin, correlating with the enhanced polymerization kinetics seen in spectroscopic data. Conversely, the weaker binding and polymerization influence of PACAP27, as noted in both DSC and anisotropy measurements, indicated limited efficacy in stabilizing actin structures. These findings highlight PACAP38 and PACAP6-38 as potent modulators of actin dynamics, potentially useful in therapeutic applications targeting cytoskeletal reorganization in neurodegenerative diseases and traumatic injuries.

Statistical analyses highlight the distinct interactions and effects of different PACAP forms on actin dynamics, with notable differences in binding and fluorescence intensities, but no significant variations in polymerization and quenching behaviors. However, PACAP isoforms had significant effects on both polymerization rates and the fluorescence quenching of G-actin.

### 3.3. The Future of In Vitro PACAP Studies

With few pioneering agents in the pharmaceutical market, PACAP shows promise as an innovative recovery agent. In vitro studies are crucial for translating compounds into drugs, and our results align with in vivo findings, suggesting a potential clinical application of PACAP due to its neuroprotective effects. PACAP38 and PACAP6-38 are notably effective, while PACAP27’s status remains uncertain. Further research could validate PACAP27’s viability. The neuroprotective and anti-apoptotic roles of PACAP isoforms warrant further research. PACAP positively influences cytoskeletal reorganization, essential for neural regeneration. In cases of axonal damage, agents enhancing filament polymerization would be a significant advancement. Like any therapeutic compound, identifying the target of PACAP—such as the polymerization block in neural regeneration—is crucial. Previous studies have shown that PACAP can reverse dystrophic structures and promote axonal growth [84,85]. PACAP38 and PACAP6-38 accelerate polymerization, suggesting that rapid administration post-injury could induce significant neuroprotective effects within 24–72 h, aiding in recovery.

Future research will encompass exploring various PACAP fragments and delving deeper into the cytoskeletal aspects of PACAP38, PACAP27, and PACAP6-38 using techniques such as TIRF microscopy, denaturation studies, co-sedimentation assays, and limited proteolysis. Investigating the interaction between PACAP and its receptor PAC1 (PAC1R) will further elucidate the molecular mechanisms underlying the effects of PACAP. Our findings underscore the potential of PACAP for clinical therapy, particularly in aiding neurological recovery, emphasizing the need to fully understand its therapeutic implications through comprehensive investigations of its interactions with PAC1.

## 4. Materials and Methods

A workflow figure (Figure 8) provides an overview of the experimental procedures.

### 4.1. Actin Preparation

Actin was prepared from acetone-dried muscle powder from rabbit skeletal muscle according to the method of Spudich and Watt [88] modified by Mossakowska et al. [89]. Shortly, the acetone-dried powder was extracted in G-buffer (4 mM TRIS (Sigma-Aldrich, Saint Louis, MO, USA), 0.1 mM CaCl_2_ (Sigma-Aldrich, Saint Louis, MO, USA), 0.2 mM ATP (VWR International LLC, Radnor, PA, USA), 0.5 mM DTT (VWR International LLC, Radnor, PA, USA), pH 7.8), mixed on ice with a magnetic stirrer (30 min, 80 rpm). The mixture was filtered through quadruple-layered gauze, supplemented with additional G-buffer, mixed further (40 min, 80 rpm), filtered again, and centrifuged (30,000 rpm, 4 °C, 30 min). The supernatant was filtered and polymerized at room temperature (20 °C) for 120 min with 50 mM KCl (Molar Chemicals Kft., Halásztelek, Hungary) and 2 mM MgCl_2_ (Sigma-Aldrich, Saint Louis, MO, USA). Solid KCl was added to achieve a final concentration of 0.6 M, followed by stirring in the refrigerator (80 rpm) until fully dissolved. After centrifugation (80,000 rpm, 30 min, 4 °C), the supernatant was removed, and G-buffer was added to the pellet, which was allowed to swell on ice for 1.5 h. The sample was homogenized and subjected to 2-day dialysis, with the G-buffer replaced the following morning and afternoon. After clarification centrifugation (80,000 rpm, 30 min, 4 °C), the supernatant was removed and actin absorbance was measured using a spectrophotometer to determine its concentration. The sample was loaded onto an S200 column (fast protein liquid chromatography, FPLC, AP Hungary Kft., Budapest, Hungary), and gel-filtered actin was run on SDS-polyacrylamide gel (SDS-PAGE), separated into unit volume fractions. Samples with high optical density were selected and centrifuged (80,000 rpm, 30 min, 4 °C). The prepared actin was labeled at Lys328 by Alexa Fluor^®^ 488 carboxylic acid succinimidyl ester (Alexa488NHS, Invitrogen, Waltham, MA, USA) for the anisotropy experiments or at Cys374 by N-(1-pyrene)iodoacetamide (pyrene, Thermo Fisher Scientific, Waltham, MA, USA) based on elaborated protocols [90,91] for the polymerization tests.

### 4.2. PACAP Synthesis and Preparation

PACAP was synthesized as previously described [92,93,94] and dissolved in G-buffer (detailed in Section 4.1) to achieve the desired concentration.

### 4.3. Steady-State Anisotropy and Fluorescence Emission Studies

For our investigation, we utilized A488NHS green-fluorescent dye, which remains stable between pH 4 and 10, with peak emission when excited at 488 nm. The NHS ester group binds to proteins. In vitro anisotropy and fluorescence emission measurements involved 0.2 µM Alexa488NHS-Mg^2+^-ATP-G-actin (Alexa488NHS labeled G-actin) with varying PACAP concentrations. Steady-state fluorescence was measured using a Horiba Jobin Yvon Fluorolog-3 spectrofluorimeter (Jobin Yvon Inc., Edison, NJ, USA), with a thermostated sample holder (HAAKE F3) and circulator (HAAKE Mess-Technik GmbH Co., Karlsruhe, Germany). The sample temperature was maintained at 20 °C. The cuvette (Hellma^®^ fluorescence, ultra Micro, Suprasil^®^ quartz, spectral range 200–2500 nm) had optical path lengths of ex = 10 mm/em = 2 mm. A488NHS-labeled actin monomers were excited at 488 nm. Sample preparation involved 50 mM potassium chloride, 1 mM magnesium chloride, 4 μM Latrunculin (LatA), 1 mM EGTA, and G-buffer, with a final volume of 150 µL. A calcium-free environment was ensured by adding 1 mM EGTA. The maximum emission wavelength was derived from the spectra. UV-VIS spectrophotometry was conducted using a JASCO V-660 spectrophotometer (JASCO International Co., Ltd., Hachioji, Tokyo, Japan) at 20 °C, with a cuvette path length of 10 mm.

For quantitative analysis, the PACAP concentration dependence of the steady-state anisotropy (r) was fitted using the following equation:(r − r_A_)/(r_AP_ − r_A_) = (A_0_ + P_0_ + K_D_ − √((A_0_ + P_0_ + K_D_)^2^ − 4 × A_0_ × P_0_))/(2 × P_0_)
where A_0_ and P_0_ represent the total G-actin and PACAP concentrations, respectively, r_A_ is the steady-state anisotropy of the PACAP-G-actin complex, r_AP_ is the steady-state anisotropy of Alexa488NHS-G-actin at a saturating amount of PACAP, and K_D_ is the dissociation equilibrium constant of the G-actin complex.

### 4.4. Actin Polymerization Assay

During the polymerization tests, we utilized a pyrene–actin environment with an actin concentration of 2.5 µM, where 5% was labeled with pyrene. Pyrene served as an external fluorophore, and its fluorescence emission was measured using a spectrofluorimeter. The excitation wavelength was 365 nm (λ_ex_ = 365 nm) and emission was measured at 407 nm (λ_em_ = 407 nm). Actin labeling required excess pyrene, along with potassium chloride (KCl) and magnesium chloride (MgCl_2_). The fluorescence emission of pyrene was measured using a Safas Xenius FLX spectrofluorimeter (SAFAS Monaco, Monaco, Monaco).

The data evaluation was based on the time versus intensity polymerization curves. The rate of polymerization was determined from the slope of the linear (elongation) phase of the pyrene polymerization curves. Data points outside the linear phase were excluded to accurately determine the slope. Linear regression was used to infer the polymerization rate. To ensure comparability, individual polymerization rates were normalized against the control, providing relative polymerization rates.

### 4.5. Fluorescence Quenching

The fluorescence quenching analysis of PACAP38, PACAP27, and PACAP6-38 was conducted using acrylamide with Horiba Jobin Yvon Fluorolog-3 fluorometer. Fluorescence emission spectra were corrected for the inner filter effect (E_corrected_) using the following equation:E_corrected_ = E × 10 ^ ((OD_ex_/2) + (OD_em_/10))
where E represents the measured fluorescence emission of G-actin (1 mg/mL), while OD_ex_ and OD_em_ denote the optical density at the excitation and emission wavelengths, respectively. The excitation wavelength was set at 295 nm and the emission was derived from integrated intensity values between 305 nm and 365 nm without acrylamide (F_0_) and with various acrylamide concentrations (F) (both excitation and emission slit widths were set at 1 nm). The dependence of fluorescence emission on acrylamide concentration ([Q]) was fitted using the Stern–Volmer equation:F_0_/F = 1 + K_SV_ × [Q] = 1 + K_q_ × τ_q_ × [Q]
where K_SV_ is the Stern–Volmer constant. Intrinsic fluorophores, such as tryptophan (Trp), tyrosine (Tyr), and phenylalanine (Phe), were identified in different PACAP forms. The relevant amino acid counts for the experiment can be found in Table 1.

### 4.6. Statistical Analysis

In this study, the averaging method was employed to present the results accurately. Final values were derived from at least three quality measurements, excluding any potentially erroneous data due to artifacts. The analyzed results were depicted in figures based on these averages, with data points represented as the mean ± standard deviation (SD), indicated by thin, vertical lines in the illustrations. For the statistical analysis, two-way ANOVA was used to assess significant differences among the PACAP forms across anisotropy, F/F_0_ fluorescence, polymerization, and fluorescence quenching measurements. When two-way ANOVA indicated significant differences, Tukey’s HSD post hoc tests were performed. Statistical significance was set at *p* < 0.05.

## 5. Conclusions

Our study yields several significant findings regarding the effects of PACAP forms on actin structure and dynamics. We discovered that PACAP38 and PACAP6-38 exhibit strong binding to G-actin, whereas PACAP27 forms a much weaker interaction. This differential binding affinity is crucial for understanding their varying impacts on cytoskeletal dynamics. Notably, PACAP6-38 significantly enhanced the polymerization rate of actin, more so than PACAP38 and PACAP27. This increase in polymerization suggests a unique role for PACAP6-38 in promoting cytoskeletal rearrangement, which could be leveraged for therapeutic purposes in neurodegenerative and traumatic injuries. Our quenching studies revealed that the PACAP forms exhibited both static and dynamic quenching properties, although no significant differences were found among the forms studied. This indicates a commonality in how these PACAP fragments interact with the quenching agents.

These findings underline the potential of PACAP38 and PACAP6-38 in clinical applications, particularly in accelerating neural regeneration and repair. Given the promising results, future research will continue to focus on these PACAP forms, exploring their mechanisms in greater detail and investigating additional fragments to enhance our understanding and therapeutic capabilities. We aim to expand on these findings by further investigating the molecular interactions of PACAP, including those with its receptor, and their therapeutic potential.

## Figures and Tables

**Figure 1 ijms-25-08063-f001:**
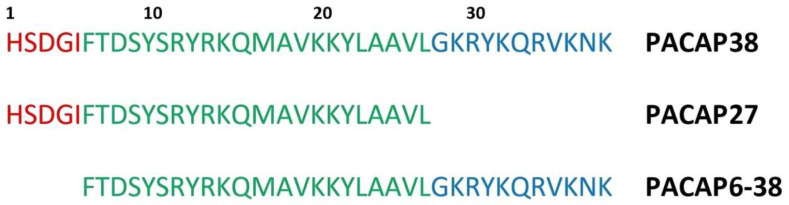
Sequences of PACAP forms (PACAP38, PACAP27, and PACAP6-38). This figure illustrates the amino acid sequences of four different forms of PACAP, highlighting the variations in their N-terminal and C-terminal regions. The numbers in the PACAP names indicate the specific amino acid orders included in each form, with the left number representing the starting amino acid from the N-terminal side and the right number indicating the ending amino acid on the C-terminal side.

**Figure 2 ijms-25-08063-f002:**
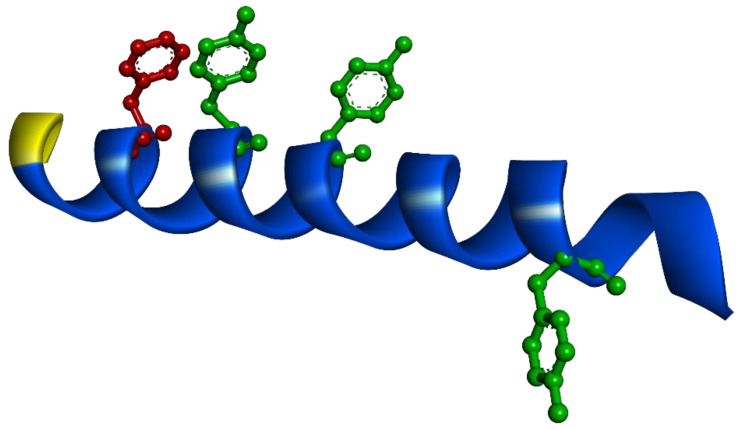
The 3D structure of PACAP27, which is identical to the first 27 residues of PACAP38. This figure is based on the PDB file 6M1I (from Cryo-EM structures of PACAP38-PAC1R-G_s_ [83]). The yellow coloring represents the N-terminal of the peptide, while the aromatic residues crucial for fluorescence are highlighted (green: tyrosine; red: phenylalanine). This figure was created using Discovery Studio Visualizer (v16.1.0.15350, Biovia).

**Figure 3 ijms-25-08063-f003:**
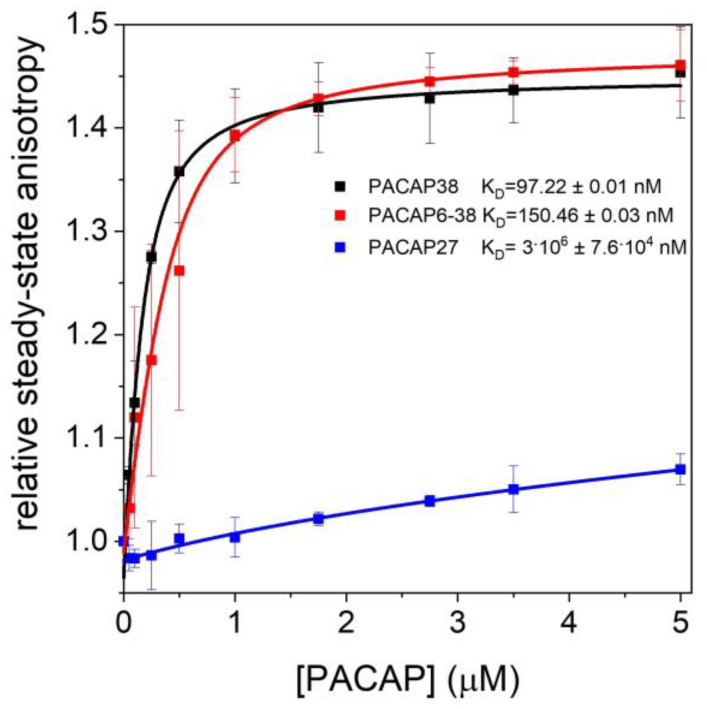
Anisotropy of 0.2 µM A488NHS-labeled G-actin as a function of increasing concentrations of PACAP isoforms (PACAP38, PACAP27, and PACAP6-38). Anisotropy measurements were conducted with PACAP concentrations ranging from 0 µM (control) to 5 µM. Data points represent the mean ± standard deviation of at least three independent experiments.

**Figure 4 ijms-25-08063-f004:**
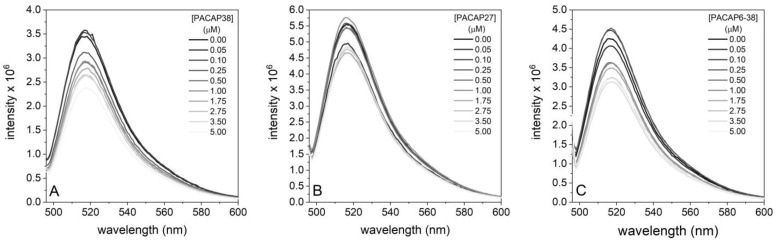
Representative emission spectra of Alexa488-labeled actin (0.2 µM) with increasing concentrations of PACAP isoforms ((**A**): PACAP38, (**B**): PACAP27, and (**C**): PACAP6-38). The spectra were recorded ranging from 0 µM (control) to 5 µM. Each spectrum illustrates changes in fluorescence intensity, with potential shifts in emission peaks, allowing for a comparative analysis of the interaction between PACAP38 and actin monomers.

**Figure 5 ijms-25-08063-f005:**
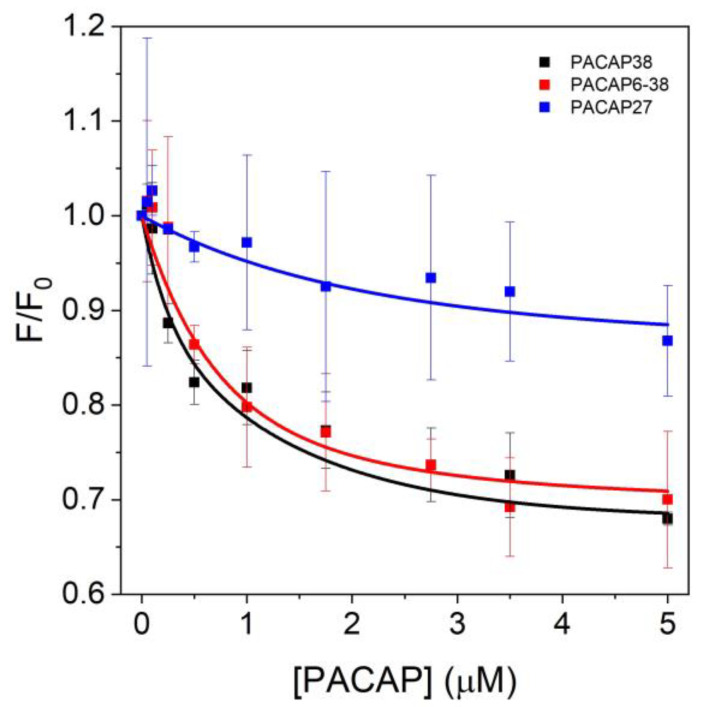
Normalized fluorescence intensity (F/F_0_) of 0.2 µM A488NHS-labeled actin as a function of increasing PACAP concentrations. Fluorescence measurements were conducted with PACAP concentrations ranging from 0 µM (control) to 5 µM. The data points represent the mean ± standard deviation from three independent experiments.

**Figure 6 ijms-25-08063-f006:**
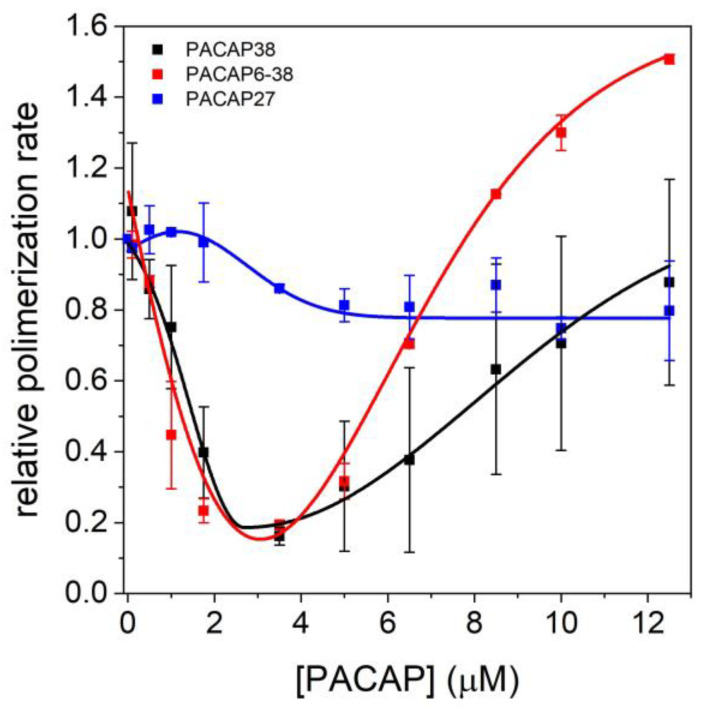
Polymerization kinetics of 2.5 µM actin (5% pyrene-labeled) as a function of increasing PACAP concentrations. The polymerization process was monitored by the increase in pyrene fluorescence over time. Polimerization assays were conducted with PACAP concentrations ranging from 0 µM (control) to 12.5 µM. Data points represent the mean ± standard deviation from three independent experiments. Two-way ANOVA did not reject the null hypothesis, indicating no significant difference between the PACAP forms.

**Figure 7 ijms-25-08063-f007:**
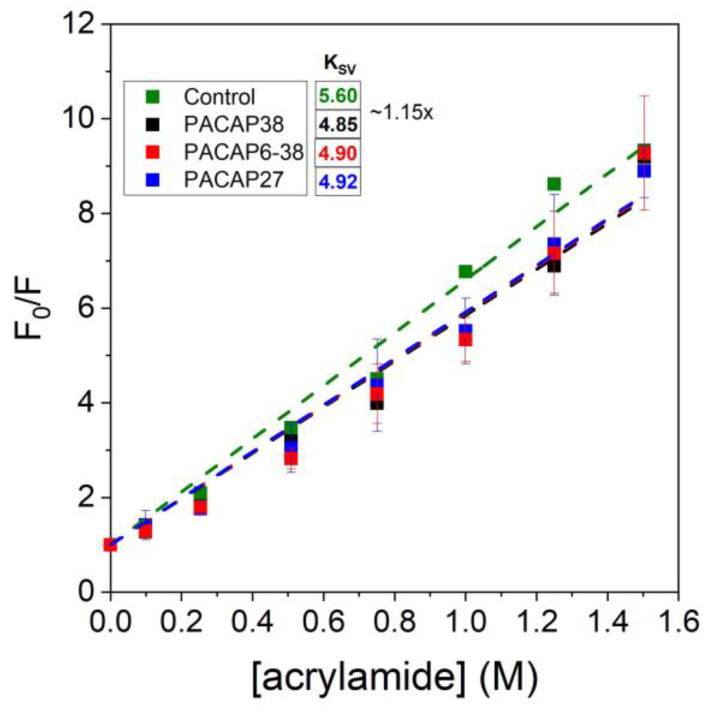
Tryptophan fluorescence quenching by acrylamide in 1 mg/mL actin as a function of various PACAP forms/fragments. The graph depicts the quenching efficiency (F_0_/F) of tryptophan fluorescence in the presence of PACAP38, PACAP27, and PACAP6-38. The quenching measurements were performed with increasing concentrations of acrylamide, and the data points represent the mean ± standard deviation from at least three independent experiments. Two-way ANOVA also did not reject the null hypothesis, indicating no significant difference between the control and PACAP isoforms.

**Figure 8 ijms-25-08063-f008:**
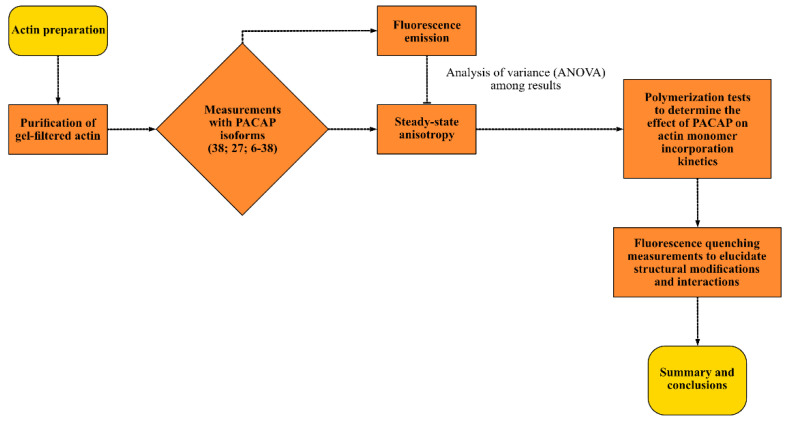
This figure outlines the sequential steps involved in the preparation, analysis, and characterization of PACAP interactions with actin. The key steps include PACAP and actin preparation, steady-state anisotropy and fluorescence emission measurements, polymerization assays, and fluorescence quenching studies.

**Table 1 ijms-25-08063-t001:** Number of intrinsic fluorophores of PACAP forms (Trp, Tyr, Phe).

Intrinsic Fluorophore	PACAP38	PACAP27	PACAP6-38
Trp	0	0	0
Tyr	4	3	4
Phe	1	1	1

## Data Availability

The original contributions presented in this study are included in the article. All relevant data are within the manuscript, and further inquiries can be directed to the corresponding author. The dataset generated and analyzed during the current study is also available from the corresponding author upon request.

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
