# Peer review of "Exploring the Role of Neuropeptide PACAP in Cytoskeletal Function Using Spectroscopic Methods"

_ijms, 2024, doi:10.3390/ijms25158063_

Round 1

Reviewer 1 Report

Comments and Suggestions for Authors

This paper demonstrates the power of fluorescence spectroscopy to elucidate the interactions of an important neuropeptide with actin. Fluorescence anisotropy probed the changes to Alexa488-actin rotational dynamics upon binding various PACAP analogues, which revealed the regions of the peptide which were crucial to binding. The fluorescence spectroscopy revealed that peptide binding was accompanied by a decrease in the quantum yield of the probe. Finally, tryptophan fluorescence quenching by acrylamide revealed that PACAP binding did influence accessibility to quencher but no evidence of gross structural changes such as unfolding. The paper is easy to read and the significance clearly explained. This work should be of interest to the neurobiology and neurochemistry community and prompts further work on how these important neuropeptides may be play a role in the cell cytoskeleton and associated activities such as locomotion or neuronal pathfinding.

Author Response

Comments 1

This paper demonstrates the power of fluorescence spectroscopy to elucidate the interactions of an important neuropeptide with actin. Fluorescence anisotropy probed the changes to Alexa488-actin rotational dynamics upon binding various PACAP analogues, which revealed the regions of the peptide which were crucial to binding. The fluorescence spectroscopy revealed that peptide binding was accompanied by a decrease in the quantum yield of the probe. Finally, tryptophan fluorescence quenching by acrylamide revealed that PACAP binding did influence accessibility to quencher but no evidence of gross structural changes such as unfolding. The paper is easy to read and the significance clearly explained. This work should be of interest to the neurobiology and neurochemistry community and prompts further work on how these important neuropeptides may be play a role in the cell cytoskeleton and associated activities such as locomotion or neuronal pathfinding.

Response 1

Dear Respected Reviewer, I would like to express my sincere gratitude for your positive and encouraging review of our manuscript. Your detailed and thoughtful comments are greatly appreciated.

We are pleased to hear that you found our study on the interactions of PACAP with actin using fluorescence spectroscopy to be compelling. Your recognition of the power of fluorescence anisotropy in elucidating the rotational dynamics of Alexa488-actin upon binding various PACAP analogues is highly valued. We are also glad that the significance of the changes in the quantum yield of the probe and the insights from tryptophan fluorescence quenching by acrylamide were clear and well-explained. Your acknowledgment of the readability of our work and its potential impact on the neurobiology and neurochemistry community is particularly gratifying. We share your enthusiasm for further research into the role of these neuropeptides in cytoskeletal functions and related activities such as locomotion and neuronal pathfinding.

Once again, thank you for your supportive review and for highlighting the significance of our findings. Your feedback is invaluable to us.

Reviewer 2 Report

Comments and Suggestions for Authors

My suggestion:

1. I suggest adding a 3D structure of PACAP peptide in the manuscript, for example by AlphaFold or Phyre2. Also, authors may highlight the important residues in the PACAP peptide, which may be crucial in the peptide functions.

2. I would also add the gene information (ADCYAP1), which encodes the PACAP peptide

3. In the methods section, a workflow figure may be useful. 

4. Since PACAP is involved in building the cytoskeleton, can it be involved in some neurodegenerative diseases? For example by misfolding or aggregation. 

5. Is there any association between PACAP and Tau protein in the cytoskeleton?

Author Response

Comments 1

I suggest adding a 3D structure of PACAP peptide in the manuscript, for example by AlphaFold or Phyre2. Also, authors may highlight the important residues in the PACAP peptide, which may be crucial in the peptide functions.

Response 1

Dear Respected Reviewer, thank you for your valuable suggestion regarding the addition of a 3D structure of the PACAP peptide to the manuscript. In response, we have generated a new figure using Discovery Studio Visualizer to illustrate the 3D structure of PACAP27, which is identical to the first 27 residues of PACAP38.

We have visualized only the N-terminal 27 residues of PACAP38 in the figure. This is because the structure of these 27 residues is well-conserved and crucial for the PAC1 receptor activation. They are clearly resolved in the structural data [1]. In contrast, the C-terminal 11 residues are not visible in the structural map due to their flexibility, and therefore, were omitted from the final reported structure. This approach ensures the accuracy and clarity of the structural representation.

The figure highlights key aromatic residues that play crucial roles in fluorescence, specifically tyrosine (green) and phenylalanine (red). We believe this visualization of the peptide's structure significantly contributes to the manuscript's clarity, thank you once again for your constructive feedback.

Comments 2

I would also add the gene information (ADCYAP1), which encodes the PACAP peptide.

Response 2

Dear Respected Reviewer, thank you for your insightful suggestion to include gene information related to PACAP. I have now added details about the ADCYAP1 gene, which encodes the PACAP peptide, to the manuscript. This addition specifies that the ADCYAP1 gene is located on chromosome 18 and produces multiple isoforms of PACAP through alternative splicing, including PACAP38 and PACAP27 [2]. This enhancement provides a clearer genetic context for the role of PACAP in the study.

Comments 3

In the methods section, a workflow figure may be useful.

Response 3

Dear Respected Reviewer, thank you for your valuable suggestion. In response, we have added a workflow figure to the Methods section of the manuscript. This figure illustrates the experimental design and procedures followed in our study, providing a clear and concise overview of our methodology.

Comments 4

Since PACAP is involved in building the cytoskeleton, can it be involved in some neurodegenerative diseases? For example, by misfolding or aggregation.

Response 4

Dear Respected Reviewer, thank you for your insightful question regarding the potential involvement of PACAP in neurodegenerative diseases. PACAP has a strong neurotrophic and neuroprotective effect, which has been investigated in many studies [3–5], it affects Schwann cell functions and myelin maturation [6], and PACAP receptors are upregulated after peripheral nerve injury [7]. PACAP promotes axonal development [8], increases axonal sprouting [9], and induces neuronal differentiation in growing and regenerated neurons [10–12].

Neuroprotective effects have been demonstrated in various neurodegenerative diseases and neuronal insults, including spinal atrophy [13], amyotrophic lateral sclerosis [14], Alzheimer's disease [15], stroke [16,17], Parkinson's disease [3,4,18,19], and Huntington chorea [20]. PACAP KO animals showed slower axonal regeneration and a more pro-inflammatory milieu, indicating that endogenous PACAP is implicated in the immune response, which is required for normal neuron regeneration following damage [10].

A recent study on PACAP and neuropathy in a streptozotocin-induced diabetes model [21] found that PACAP reduced morphological signs of nerve injury (axon-myelin separation, elevated mitochondrial number in myelinated axons, unmyelinated fiber atrophy, and basement membrane thickening of endoneurial vessels) while also attenuating pressure and touch hyperalgesia. Furthermore, the pain processing centers (dorsal spinal horn and periaqueductal grey matter) exhibited lower FosB immunoreactivity. The gene encoding PACAP was most substantially increased in carpal tunnel syndrome patients, and its expression was associated with intraepidermal nerve fiber recovery [11].

Several animal and human studies have reported reduced PACAP levels, altered PACAP receptor expression, and the beneficial effect of PACAP on cognition and memory in conditions associated with cognitive deficits such as aging, Alzheimer's, Parkinson's and Huntington's disease, schizophrenia, and fragile X syndrome. Motor symptoms take years to manifest, whereas non-motor symptoms are frequently linked to Parkinson's disease. According to Braak's theory, Lewy bodies, or alpha-synuclein inclusions, first develop in the peripheral nervous system before moving within and impacting an increasing number of locations in the central nervous system [22]. Neurons in the olfactory bulb and enteric nervous system, which are in charge of non-motor symptoms, are involved prior to the degeneration of the substantia nigra, and levodopa medication does not always make these dysfunctions better. Hyposmia/anosmia and constipation are two of the earliest non-motor symptoms of Lewy body deposits, which first arise in the olfactory bulb, enteric nervous system, and dorsal motor nucleus of the vagus [22,23].

The studies presented above indicate that the PACAP is a key regulator of neuronal regeneration. We have incorporated the essential part of the above information into the manuscript.

Comments 5

Is there any association between PACAP and Tau protein in the cytoskeleton?

Response 5

Dear Respected Reviewer, thank you for your constructive suggestion regarding the interaction between PACAP and Tau protein. We have incorporated the relevant information into the manuscript. Specifically, we added details about how PACAP influences Tau phosphorylation and mitigates harmful Tau aggregates, which is crucial for maintaining cytoskeletal integrity and protecting against neurodegenerative diseases such as Alzheimer's disease. The revised text now explains PACAP's role in increasing Tau phosphorylation, counteracting the dephosphorylation effects of C2-ceramide, and its protective role against cytoskeletal degradation in neurons by modulating phosphorylation and proteolysis of Tau [24].

PACAP can mitigate neuronal damage and promote cell survival. However, while PACAP itself does not typically misfold or aggregate, its regulatory role in the cytoskeleton and interaction with proteins such as Tau could influence the pathology of neurodegenerative diseases. Misfolding and aggregation of proteins like Tau are hallmarks of diseases such as Alzheimer's [15]. PACAP's ability to increase Tau phosphorylation and prevent its dephosphorylation suggests that it could play a protective role against the formation of neurofibrillary tangles, which are characteristic of Alzheimer's disease. Furthermore, PACAP has been shown to inhibit caspase-3 activity, reducing Tau cleavage and potentially mitigating Tau aggregation and its associated cytoskeletal disruptions [25,26]. Therefore, while PACAP itself may not misfold or aggregate, its modulatory effects on Tau and other cytoskeletal components suggest a significant role in protecting against neurodegenerative processes associated with protein misfolding and aggregation  [15,24–27]. We have incorporated this information into the manuscript.

References

  1. Wang, J.; Song, X.; Zhang, D.; Chen, X.; Li, X.; Sun, Y.; Li, C.; Song, Y.; Ding, Y.; Ren, R.; et al. Cryo-EM Structures of PAC1 Receptor Reveal Ligand Binding Mechanism. Cell Res 2020, 30, 436–445, doi:10.1038/s41422-020-0280-2.
  2. Freson, K.; Hashimoto, H.; Thys, C.; Wittevrongel, C.; Danloy, S.; Morita, Y.; Shintani, N.; Tomiyama, Y.; Vermylen, J.; Hoylaerts, M.F.; et al. The Pituitary Adenylate Cyclase–Activating Polypeptide Is a Physiological Inhibitor of Platelet Activation. J Clin Invest 2004, 113, 905–912, doi:10.1172/JCI19252.
  3. Reglodi, D.; Atlasz, T.; Szabo, E.; Jungling, A.; Tamas, A.; Juhasz, T.; Fulop, B.D.; Bardosi, A. PACAP Deficiency as a Model of Aging. Geroscience 2018, 40, 437–452, doi:10.1007/s11357-018-0045-8.
  4. Reglodi, D.; Tamas, A.; Jungling, A.; Vaczy, A.; Rivnyak, A.; Fulop, B.D.; Szabo, E.; Lubics, A.; Atlasz, T. Protective Effects of Pituitary Adenylate Cyclase Activating Polypeptide against Neurotoxic Agents. Neurotoxicology 2018, 66, 185–194, doi:10.1016/j.neuro.2018.03.010.
  5. Lee, E.H.; Seo, S.R. Neuroprotective Roles of Pituitary Adenylate Cyclase-Activating Polypeptide in Neurodegenerative Diseases. BMB Rep 2014, 47, 369–375, doi:10.5483/BMBRep.2014.47.7.086.
  6. Maugeri, G.; D’Amico, A.G.; Musumeci, G.; Reglodi, D.; D’Agata, V. Effects of PACAP on Schwann Cells: Focus on Nerve Injury. Int J Mol Sci 2020, 21, 8233, doi:10.3390/ijms21218233.
  7. Woodley, P.K.; Min, Q.; Li, Y.; Mulvey, N.F.; Parkinson, D.B.; Dun, X. Distinct VIP and PACAP Functions in the Distal Nerve Stump During Peripheral Nerve Regeneration. Front Neurosci 2019, 13, 1326, doi:10.3389/fnins.2019.01326.
  8. Fukiage, C.; Nakajima, T.; Takayama, Y.; Minagawa, Y.; Shearer, T.R.; Azuma, M. PACAP Induces Neurite Outgrowth in Cultured Trigeminal Ganglion Cells and Recovery of Corneal Sensitivity after Flap Surgery in Rabbits. Am J Ophthalmol 2007, 143, 255–262, doi:10.1016/j.ajo.2006.10.034.
  9. Suarez, V.; Guntinas-Lichius, O.; Streppel, M.; Ingorokva, S.; Grosheva, M.; Neiss, W.F.; Angelov, D.N.; Klimaschewski, L. The Axotomy-Induced Neuropeptides Galanin and Pituitary Adenylate Cyclase-Activating Peptide Promote Axonal Sprouting of Primary Afferent and Cranial Motor Neurones. European Journal of Neuroscience 2006, 24, 1555–1564, doi:10.1111/j.1460-9568.2006.05029.x.
  10. Armstrong, B.; Abad, C.; Chhith, S.; Cheung-Lau, G.; Hajji, O.; Nobuta, H.; Waschek, J. Impaired Nerve Regeneration and Enhanced Neuroinflammatory Response in Mice Lacking Pituitary Adenylyl Cyclase Activating Peptide (PACAP). Neuroscience 2008, 151, 63–73, doi:10.1016/j.neuroscience.2007.09.084.
  11. Baskozos, G.; Sandy-Hindmarch, O.; Clark, A.J.; Windsor, K.; Karlsson, P.; Weir, G.A.; McDermott, L.A.; Burchall, J.; Wiberg, A.; Furniss, D.; et al. Molecular and Cellular Correlates of Human Nerve Regeneration: ADCYAP1/PACAP Enhance Nerve Outgrowth. Brain 2020, 143, 2009–2026, doi:10.1093/brain/awaa163.
  12. Tsuchida, M.; Nakamachi, T.; Sugiyama, K.; Tsuchikawa, D.; Watanabe, J.; Hori, M.; Yoshikawa, A.; Imai, N.; Kagami, N.; Matkovits, A.; et al. PACAP Stimulates Functional Recovery after Spinal Cord Injury through Axonal Regeneration. J Mol Neurosci 2014, 54, 380–387, doi:10.1007/s12031-014-0338-z.
  13. Martínez-Rojas, V.A.; Jiménez-Garduño, A.M.; Michelatti, D.; Tosatto, L.; Marchioretto, M.; Arosio, D.; Basso, M.; Pennuto, M.; Musio, C. ClC-2-like Chloride Current Alterations in a Cell Model of Spinal and Bulbar Muscular Atrophy, a Polyglutamine Disease. J Mol Neurosci 2021, 71, 662–674, doi:10.1007/s12031-020-01687-5.
  14. Toth, D.; Szabo, E.; Tamas, A.; Juhasz, T.; Horvath, G.; Fabian, E.; Opper, B.; Szabo, D.; Maugeri, G.; D’Amico, A.G.; et al. Protective Effects of PACAP in Peripheral Organs. Front Endocrinol (Lausanne) 2020, 11, 377, doi:10.3389/fendo.2020.00377.
  15. Chen, X.-Y.; Du, Y.-F.; Chen, L. Neuropeptides Exert Neuroprotective Effects in Alzheimer’s Disease. Front. Mol. Neurosci. 2019, 11, doi:10.3389/fnmol.2018.00493.
  16. Nonaka, N.; Banks, W.A.; Shioda, S. Pituitary Adenylate Cyclase-Activating Polypeptide: Protective Effects in Stroke and Dementia. Peptides 2020, 130, 170332, doi:10.1016/j.peptides.2020.170332.
  17. Cherait, A.; Maucotel, J.; Lefranc, B.; Leprince, J.; Vaudry, D. Intranasal Administration of PACAP Is an Efficient Delivery Route to Reduce Infarct Volume and Promote Functional Recovery After Transient and Permanent Middle Cerebral Artery Occlusion. Front Endocrinol (Lausanne) 2021, 11, 585082, doi:10.3389/fendo.2020.585082.
  18. Jungling, A.; Reglodi, D.; Maasz, G.; Zrinyi, Z.; Schmidt, J.; Rivnyak, A.; Horvath, G.; Pirger, Z.; Tamas, A. Alterations of Nigral Dopamine Levels in Parkinson’s Disease after Environmental Enrichment and PACAP Treatment in Aging Rats. Life (Basel) 2021, 11, 35, doi:10.3390/life11010035.
  19. Pham, D.; Polgar, B.; Toth, T.; Jungling, A.; Kovacs, N.; Balas, I.; Pal, E.; Szabo, D.; Fulop, B.D.; Reglodi, D.; et al. Examination of Pituitary Adenylate Cyclase-Activating Polypeptide in Parkinson’s Disease Focusing on Correlations with Motor Symptoms. GeroScience 2022, 44, 785–803, doi:10.1007/s11357-022-00530-6.
  20. Tamás, A.; Lubics, A.; Lengvári, I.; Reglódi, D. Protective Effects of PACAP in Excitotoxic Striatal Lesion. Ann N Y Acad Sci 2006, 1070, 570–574, doi:10.1196/annals.1317.083.
  21. Kiss, P.; Banki, E.; Gaszner, B.; Nagy, D.; Helyes, Z.; Pal, E.; Reman, G.; Toth, G.; Tamas, A.; Reglodi, D. Protective Effects of PACAP in a Rat Model of Diabetic Neuropathy. International Journal of Molecular Sciences 2021, 22, 10691, doi:10.3390/ijms221910691.
  22. Cersosimo, M.G.; Benarroch, E.E. Pathological Correlates of Gastrointestinal Dysfunction in Parkinson’s Disease. Neurobiol Dis 2012, 46, 559–564, doi:10.1016/j.nbd.2011.10.014.
  23. Yan, F.; Chen, Y.; Li, M.; Wang, Y.; Zhang, W.; Chen, X.; Ye, Q. Gastrointestinal Nervous System α-Synuclein as a Potential Biomarker of Parkinson Disease. Medicine (Baltimore) 2018, 97, e11337, doi:10.1097/MD.0000000000011337.
  24. Falluel-Morel, A.; Vaudry, D.; Aubert, N.; Galas, L.; Benard, M.; Basille, M.; Fontaine, M.; Fournier, A.; Vaudry, H.; Gonzalez, B.J. PACAP and Ceramides Exert Opposite Effects on Migration, Neurite Outgrowth, and Cytoskeleton Remodeling. Ann N Y Acad Sci 2006, 1070, 265–270, doi:10.1196/annals.1317.024.
  25. Arimura, A. Perspectives on Pituitary Adenylate Cyclase Activating Polypeptide (PACAP) in the Neuroendocrine, Endocrine, and Nervous Systems. Jpn J Physiol 1998, 48, 301–331, doi:10.2170/jjphysiol.48.301.
  26. Gonzalez, B.J.; Vaudry, D.; Basille, M.; Rousselle, C.; Falluel-Morel, A.; Vaudry, H. Function of PACAP in the Central Nervous System. In Pituitary Adenylate Cyclase-Activating Polypeptide; Vaudry, H., Arimura, A., Eds.; Springer US: Boston, MA, 2003; pp. 125–151 ISBN 978-1-4615-0243-2.
  27. Manecka, D.-L.; Boukhzar, L.; Falluel-Morel, A.; Lihrmann, I.; Anouar, Y. PACAP Signaling in Neuroprotection. In Pituitary Adenylate Cyclase Activating Polypeptide — PACAP; Reglodi, D., Tamas, A., Eds.; Springer International Publishing: Cham, 2016; pp. 549–561 ISBN 978-3-319-35135-3.

Round 2

Reviewer 2 Report

Comments and Suggestions for Authors

The authors fulfilled my suggestions. Thank you.